# Refinement of *Leishmania donovani* Genome Annotations in the Light of Ribosome-Protected mRNAs Fragments (Ribo-Seq Data)

**DOI:** 10.3390/genes14081637

**Published:** 2023-08-17

**Authors:** Alejandro Sánchez-Salvador, Sandra González-de la Fuente, Begoña Aguado, Phillip A. Yates, Jose M. Requena

**Affiliations:** 1Centro de Biología Molecular Severo Ochoa (CSIC-UAM), Departamento de Biología Molecular, Instituto Universitario de Biología Molecular (IUBM), Universidad Autónoma de Madrid, 28049 Madrid, Spain; alejandro.sanchez@cbm.csic.es; 2Centro de Biología Molecular Severo Ochoa (CSIC-UAM), Genomic and NGS Facility (GENGS), 28049 Madrid, Spain; sandra.g@cbm.csic.es (S.G.-d.l.F.); baguado@cbm.csic.es (B.A.); 3Department of Chemical Physiology & Biochemistry, Oregon Health & Science University, Portland, OR 97239, USA; 4Centro de Investigación Biomédica en Red (CIBERINFEC), Instituto de Salud Carlos III, 28029 Madrid, Spain

**Keywords:** *Leishmania*, ribosome profiling, Ribo-seq, uORFs, genome, transcriptome

## Abstract

Advances in next-generation sequencing methodologies have facilitated the assembly of an ever-increasing number of genomes. Gene annotations are typically conducted via specialized software, but the most accurate results require additional manual curation that incorporates insights derived from functional and bioinformatic analyses (e.g., transcriptomics, proteomics, and phylogenetics). In this study, we improved the annotation of the *Leishmania donovani* (strain HU3) genome using publicly available data from the deep sequencing of ribosome-protected mRNA fragments (Ribo-Seq). As a result of this analysis, we uncovered 70 previously non-annotated protein-coding genes and improved the annotation of around 600 genes. Additionally, we present evidence for small upstream open reading frames (uORFs) in a significant number of transcripts, indicating their potential role in the translational regulation of gene expression. The bioinformatics pipelines developed for these analyses can be used to improve the genome annotations of other organisms for which Ribo-Seq data are available. The improvements provided by these studies will bring us closer to the ultimate goal of a complete and accurately annotated *L. donovani* genome and will enhance future transcriptomics, proteomics, and genetics studies.

## 1. Introduction

Leishmaniasis refers to the spectrum of diseases caused by several species of the genus *Leishmania*. Leishmaniasis ranks second after malaria among parasitic diseases, with an annual incidence of approximately 0.2 to 0.4 million cases of visceral leishmaniasis (VL) and 0.7 to 1.2 million cases of cutaneous leishmaniasis (CL) worldwide [1,2]. The visceral form of the disease is fatal if left untreated and is caused by *Leishmania donovani* and *Leishmania infantum*, the latter of which also affects dogs [3].

Given the medical relevance of *Leishmania* parasites, substantial efforts have been made to determine their genome sequences in order to understand their molecular biology and to develop strategies to combat them. Classic Sanger sequencing methods were used to determine the genome sequence of *Leishmania major* (the main species causing CL), which was published in 2005 [4]. While this resulted in a high-quality genome assembly that still serves as the reference *L. major* genome, a recent reassembly based on a combination of second and third next-generation sequencing (NGS) methodologies has introduced some improvements [5]. In the last decade, the extraordinary progress in NGS technologies, combined with a significant reduction in sequencing costs, has enabled the determination of genome sequences for many *Leishmania* species (most of them are available in the TriTrypDB repository [6]). However, the gene annotation of *Leishmania* genomes is a difficult and challenging task because of the early evolutionary branching of these organisms from the main trunk of the eukaryotic tree [7]. Thus, for a significant number of predicted genes (around 40%), the encoded proteins are categorized as hypothetical proteins due to a lack of significant sequence homology with proteins from model eukaryotic organisms. Moreover, the low level of sequence conservation in those cases in which orthologues are identified in model organisms makes it difficult to predict with certainty the initiating methionine in *Leishmania* open reading frames (ORFs) based on their homology with orthologous sequences. Consequently, the automatic annotation of *Leishmania* genomes, usually carried out via the Companion server [8], should not be considered definitive unless the predictions are supported by additional bioinformatic or experimental evidence. For example, the delineation of *Leishmania* transcriptomes has facilitated the revision of the translation start sites for hundreds of ORFs in which the predicted start site was found to exceed the transcript 5′-end boundaries [5,9,10,11]. Similarly, an analysis of proteomic data via a proteogenomics approach uncovered new protein-coding genes and extended the 5′-ends of several ORFs which were initially predicted to be shorter via automated annotation [12].

The sequencing of ribosome-protected mRNA fragments (AKA, ribosome profiling or Ribo-Seq) allows for the detection of those transcripts that are actively being translated at a given time in a cell under defined conditions [13]. The combination of the nuclease digestion of ribosome–mRNA complexes (a translating ribosome typically protects a 29–30-nucleotide length of the mRNA from the nucleases’ action) and the deep sequencing of the protected mRNA fragments (i.e., ribosome footprints) highlights which portions of a given transcript are occupied by ribosomes and are thus undergoing translation. Hence, Ribo-Seq data can enhance genome annotation by detecting novel ORFs and accurately identifying the initiation codon for translated ORFs [13].

Ribosome profiling studies have been conducted in the related kinetoplastid parasites *Trypanosoma cruzi* [14] and *Trypanosoma brucei* [15,16,17]. In *T. brucei*, the Ribo-Seq data were used to curate genome annotations; in particular, 225 previously unannotated coding sequences were uncovered, and the proper start codon was found to be misannotated in more than 400 ORFs [15,17]. In both parasites, Ribo-Seq data also served to define short upstream open reading frames (uORFs) in the 5′ UTRs of several mRNAs [16]. Recently, the Clos laboratory generated a ribosome footprint dataset from *L. donovani* promastigotes as part of a study examining the role of HSP90 in regulating gene expression [18]. In this work, we reutilized these Ribo-Seq data to improve the annotations for a significant number of protein-coding genes in the *L. donovani* (HU3 strain) genome, which was previously assembled and annotated by our group from a combination of short- and long-read sequencing data [11]. Like the other kinetoplastid Ribo-Seq analyses, we identified uORFs in the 5′ UTRs of a subset of *L. donovani* mRNAs, suggesting a potential conserved role for uORFs in the translational regulation of kinetoplastids.

## 2. Materials and Methods

### 2.1. Ribosome Footprint Data

A detailed description of the samples and footprint methodology were published elsewhere [18]. Briefly, promastigotes of the *L. donovani* strain 1SR (MHOM/SD/62/1SR) were cultured at 25 °C in supplemented Medium 199 (Sigma-Aldrich) and harvested at a cell density of 6.7 × 10^6^ mL^−1^. After the samples underwent cellular lysis and RNase I-treatment, isolated ribosomes (monosomes) were purified via sucrose gradient (10% to 50% (wt/vol)) centrifugation. Footprint RNA from the monosome fraction was isolated and used for the construction of libraries, following the protocol of Ingolia et al. [13] with minor modifications. Finally, the libraries were then sequenced using an Illumina NextSeq 500 system. Raw sequencing reads (Illumina single-end reads in FASTQ format) were downloaded at the NCBI Sequence Read Archive (SRA) under project number PRJNA495919. In particular, the 24,741,340 (1 × 76 nucleotides) Ribo-Seq reads from sample SRR8040414 (derived from *L. donovani* promastigotes and used as the control sample) were used.

### 2.2. Processing and Mapping of Reads

The Trimmomatic tool [19] was used to clip adapters. Additionally, rRNA-containing reads were filtered after they were aligned against the rRNA sequences existing in the *L. donovani* genome by using the BWA-MEM aligner [20], according the sqtk3 pipeline (https://github.com/lh3/seqtk; accessed on 11 August 2023). The *L. donovani* genome used as reference is available from the ENA (SAMEA104389403), Leish-ESP (http://leish-esp.cbm.uam.es/; accessed on 11 August 2023), and/or Mendeley dataset (https://data.mendeley.com/datasets/b82fm2w2h9/2; accessed on 11 August 2023) repositories. Thereafter, the BWA-MEM aligner was also used to map the remaining reads to the genome, and the resulting alignment files were sorted and indexed using SAMtools [21]. The results are available in the following Mendeley dataset: https://data.mendeley.com/datasets/cgptcyhvd5/1 (accessed on 11 August 2023).

The tool htseq-count [22,23] was used to count the reads mapping to every transcript and the counts mapping at coding regions (CDSs) [--mode union] and the 5′- and 3′-unstranlated regions (5′-UTR and 3′-UTR) [--mode intersection-strict] within a transcript. Overlapping reads on a 5′-UTR and a CDS or a CDS and a 3′-UTR were assigned to the CDS. Ambiguous reads (mapping to two or more genomic regions) were mapped to all possible regions and counted accordingly [--nonunique all].

### 2.3. Preparation of a Database for All Possible ORFs

Two Python scripts were developed to extract all the possible ORFs (longer than 60 triplets), considering all six reading frames, from any given genome sequence. Both scripts are publicly available in our GitHub repository (https://github.com/CBMSO-L302/Genomic-Tools; accessed on 11 August 2023). These scripts use a nucleotide genomic FASTA file as input. For this work, we used the *L. donovani* HU3 genome sequence [11], which can be accessed via the following link: https://data.mendeley.com/datasets/b82fm2w2h9/2 (accessed on 11 August 2023). As output, the script coord_prot_to_gff.py creates a gff file with the coordinates for all possible ORFs (also indicating the strand, + or −), whereas the script DNA_to_proteins.py generates a FASTA file containing the amino acid sequences of the possible ORFs. The resulting files can be downloaded through the following link: https://data.mendeley.com/datasets/6b54424fgs/1 (accessed on 11 August 2023).

### 2.4. Identification of Ribosome-Protected uORFs

A database consisting of all the possible ORFs (between 11 and 100 triplets) was generated from the *L. donovani* HU3 genome sequence via an in-house Python script (publicly available on https://github.com/CBMSO-L302/Genomic-Tools; accessed on 11 August 2023). Subsequently, the 5′-UTR coordinates of each gene were extracted and processed using bedtools software [24] to identify those ORFs from the database that fit into the 5′-UTR and were located on the same strand of the corresponding main CDS. Finally, only those ORFs covered by six or more Ribo-Seq reads were labelled as putative uORFs.

### 2.5. BLAST Searches Using the TriTryDB Repository

The TriTryDB resource (https://tritrypdb.org; accessed on 11 August 2023), included in the VEuPathDB bioinformatics resource center [25], was used to analyze the orthology and synteny gene groups among the trypanosomatids. Also, in this repository, the BLAST tool was tremendously useful for the manual curation of misannotated genes and for mining the sequences from a large collection of *Leishmania* genomes.

### 2.6. Data Availability

The analyses described in this work have resulted in the correction of a significant number of protein sequences annotated from the *L. donovani* (HU3) genome. This information is being deposited at EMBL-EBI/NCBI, and it will appear as an updated version of the entry ERS2007345|SAMEA104389403. Also, in the Appendix A, we provide the improved annotations of all genes identified to date in the *L. donovani* (HU3) genome. Additionally, a FASTA file with the improved proteome annotations may be downloaded through the following Mendeley dataset: https://data.mendeley.com/datasets/mcm8wyr2k4/1 (accessed on 11 August 2023).

## 3. Results and Discussion

In 2018, Bifeld et al. [18] published a study in which ribosome footprint (Ribo-Seq) data were generated from *L. donovani* promastigote samples. At that time, the available genome assembly for *L. donovani* was quite incomplete and fragmented, and the authors opted to align the resulting sequence reads to the *L. infantum* reference genome, which has a remarkable sequence conservation with the *L. donovani* genome [26]. The Ribo-Seq study by Bifeld assessed the effects of HSP90 inhibition on *Leishmania* protein synthesis, comparing ribosome footprint data from inhibitor-treated and untreated control samples. In our studies, to avoid any unforeseen alterations in ribosome footprints linked to the inhibition of HSP90 (a protein which has pleiotropic effects in the cell), we analyzed only those datasets derived from control samples.

The Ribo-Seq reads were mapped to the *L. donovani* genome (HU3 strain) [11]. As was the case with the original mapping of the data to the *L. infantum* genome by Bifeld and colleagues [18], the footprints faithfully mapped to the predicted ORFs. However, a detailed inspection of the data indicated that a significant number of Ribo-Seq reads mapped either on putative 5′-UTRs or on RNAs categorized as non-coding. It should be noted that 2301 SL-containing transcripts were annotated as non-coding RNAs (ncRNAs) in the *L. donovani* (HU3 strain) transcriptome [11].

To uncover possible protein-coding genes among the ncRNAs, Ribo-Seq reads mapping on those ncRNAs were counted. Those ncRNAs with 20 or more mapped Ribo-Seq reads were selected for further analysis. To facilitate the identification of possible coding sequences (CDSs), a database consisting of all possible ORFs longer than 20 triplets (i.e., codons) was created via the virtual translation of all six possible reading frames of the *L. donovani* (HU3) genome sequence (https://data.mendeley.com/datasets/6b54424fgs/1; accessed on 11 August 2023). This file (gff format) was visualized using the IGV viewer tool [27] together with the Ribo-Seq reads aligned to the *L. donovani* genome (https://data.mendeley.com/datasets/cgptcyhvd5/1; accessed on 11 August 2023). Those transcripts in which Ribo-Seq reads mapped evenly on a possible ORF were postulated to be potential protein-coding genes. In all, 71 transcripts were postulated to have protein-coding functions (see Appendix A). Figure 1 illustrates the process followed for the identification of plausible new protein-coding transcripts. A hypothetical protein measuring 127 amino acids in length was postulated to be encoded in transcript LDHU3_36.4050T (previously annotated as ncRNA). A search in the TriTrypDB repository for orthologous proteins annotated in the genomes of another *Leishmania* species showed that N-terminal-truncated, highly conserved homologous proteins were annotated in *Leishmania turanica* (LTULEM423_360036100), *Leishmania arabica* (LARLEM1108_360036300), and *Leishmania braziliensis* (LBRM2903_350038200), among others; however, no orthologues were annotated in the genomes of other prototypical species such as *L. major*, *L. infantum*, and *L. mexicana*. A BLAST search using the LDHU3_36.4050 (new CDS; Figure 1) ORF against the *L. major* transcriptome [5] showed that a highly conserved sequence exists in the transcript LMJFC_360040500-T, which is currently annotated as an ncRNA [5]. The translation of the sequence yielded a polypeptide very similar in length and sequence to the one annotated in *Leishmania orientalis* (see Figure 1). Taken together, these data support the conclusion that LDHU3_36.4050 is a protein-coding gene that was overlooked during the automatic annotation process carried out using the Companion tool [11].

Another example of a new protein-coding gene is shown in Figure 2. The Ribo-Seq reads mapped to a 237-amino acid ORF on transcript LDHU3_12.0020T, which was previously annotated as an ncRNA. A search using the TriTrypDB BLAST tool indicated that genes coding for orthologous proteins are annotated in a few *Leishmania* genomes (see Figure 2B, and its caption) but are absent from most *Leishmania* genomes deposited in this repository. To determine whether the absence of orthologous genes in many *Leishmania* genomes was due to a failure in the annotation process, as occurred during the annotation of the *L. donovani* (HU3) genome, we searched the *L. mexicana* (U1103 strain) genome sequence using the LDHU3_12.0020 ORF as a query. The BLAST algorithm identified a perfect match (no gaps, 86% sequence identity, and an e-value = 0.0) at positions 2252–2965 of the *L. mexicana* chromosome 12 (it should be noted that the LDHU3_12.0020 ORF is located in an equivalent region, i.e., on the 5′-end of *L. donovani* chromosome 12, see Figure 2). The encoded protein is also 237 amino acids in length, and it is almost identical to that annotated in *L. major* (LV39 strain) and *L. donovani* (this work; Figure 2B). Therefore, we can conclude that LDHU3_12.0020 is surely a protein-coding gene and that orthologous proteins are also encoded in most (or possibly all) *Leishmania* genomes, even though orthologous genes are missed in the current annotations of many *Leishmania* genomes.

Next, the distribution of the Ribo-Seq reads on protein-coding genes were mapped onto the 5′-UTR, CDS, or 3′-UTR segments of the genes. As expected, almost no reads (three or less) mapped on the 3′-UTRs; however, for around two thousand genes, a significant number of reads mapped to the 5′-UTRs. These results support the possibility that some of the annotated CDSs might be extended at their 5′-ends. Using the all-possible-ORFs database (see Section 2), the analysis was automated to extend CDSs at their 5′-ends when a possible larger ORF existed; extensions that surpassed the transcript boundaries were avoided. The Ribo-Seq reads were then counted on the expanded CDS. When an increase of 20 or more reads occurred for the new theoretical CDS relative to the original annotated CDS, a manual inspection was performed. In total, our analyses concluded that 588 genes were likely misannotated and that the 5′-ends of their CDSs should be expanded (the Appendix A contain a list of these genes).

Figure 3 shows an example in which the extension of the CDS was suggested by the mapping of Ribo-Seq reads and supported by comparative genomics with other *Leishmania* species. LDHU3_17.0810 is currently annotated at TriTryDB.org as coding for a hypothetical protein (but is conserved among trypanosomatids) that is 2500 amino acids in length. However, the presence of a significant number of Ribo-Seq reads evenly distributed on the 5′-UTR was suggestive of a misannotation of the CDS. In fact, there was a theoretical ORF that fit perfectly in the genomic region covered by the mapped Ribo-Seq reads (Figure 3A). The protein encoded by this new CDS would be 3702 amino acids in length. A BLAST search using the new postulated protein sequence against predicted proteomes for the *Leishmania* species deposited in TriTrypDB (release 62) showed that indeed, orthologous proteins of a similar size (around 3700 amino acids) to the new predicted sequence for LDHU3_17.0810 are currently annotated in most *Leishmania* proteomes (see Figure 3 for two examples). The reason for the previous misannotation was unclear, but this finding highlights the limitations and risks of relying solely on automatic annotations. In the predicted proteomes for *L. infantum* (strain JPCM5), *L. donovani* (strain CL-SL), *Leishmania panamensis* (strain PSC-1), *L. amazonensis* (strain M2269), and *L. mexicana* (strain U1103), the orthologous proteins are 150 amino acids shorter than those predicted from the re-annotated LDHU3_17.0810 CDS. Therefore, before working with the orthologous proteins from these species, a possible reannotation should be considered.

LDHU3_08.1530 serves as a second example of a gene misannotated with a shortened CDS (Figure 4). The current annotation indicates that this gene codes for a polypeptide of 442 amino acids. However, after visualizing the Ribo-Seq reads mapping on the gene, a 5′ extension of the CDS was postulated. In fact, the ribosome-protected region of the transcript corresponded perfectly with a longer ORF that would encode a 555-amino acid protein. Interestingly, a BLAST search using the TriTrypDB repository showed that a full-length orthologous protein (555 amino acids in length) is annotated in the *L. aethiopica* (L147) and *L. major* (Friedlin) genomes (Figure 4), among others. However, it is likely that the shortened versions of the corresponding ORFs in the *L. infantum*, *L. donovani* and *L. mexicana* genomes are misannotated and should be extended to match our updated LDHU3_08.1530 ORF.

The alignment of the ribosome footprints to the genome revealed eight cases in which a single long transcript had been misannotated as two separate transcripts. An example is shown in Figure 5 and involves the transcripts LDHU3_12.T0820 and LDHU3_12.T0830 (according to previous annotations). Previously, the transcript LDHU3_12.T0820 was annotated as an ncRNA, but several spliced leader (SL) addition sites were mapped at its 5′-end. whereas transcript LDHU3_12.0830 was annotated as protein-coding gene; however, no SL-addition sites were found. The transcriptome was assembled via the Cufflinks tool, using the Illumina RNA-seq data and default parameters as described elsewhere [11]. Artificial discontinuities in a transcript may arise when low RNA-seq coverage causes the transcript to be split into two moieties. However, the distribution of Ribo-Seq reads on this genomic region, together with the fact that an ORF spanning both transcripts is possible, suggested that both transcripts should be fused into one. We maintained the ID LDHU3_12.0830T for the new transcript as it represents a 5′-end extension of the former one (Figure 5). Now, this transcript would encode for a protein of 2325 amino acids, while the previously annotated one had only 1587 amino acids. In support of this reannotation is the finding that the complete form of the protein is annotated in the genomes of most *Leishmania* species deposited in TriTryDB (Figure 5B shows two examples).

Another example in which transcript and ORF misannotations were resolved using Ribo-Seq data is shown in Figure 6. In the chromosome 17 region depicted in panel A, based on the presence of SL-addition sites, the annotation pipeline split up the primary assembled transcript into two transcripts, LDHU3_17.1520 (annotated as an ncRNA) and LDHU3_17.1530 (coding for a hypothetical protein of 1976 amino acids). Nevertheless, the mapping of the Ribo-Seq reads to the genome suggested that both transcripts are being translated in a comparable manner, and more importantly, the Ribo-Seq reads mapped uniformly onto a larger ORF, which extends through the genomic region corresponding to transcript LDHU3_17.1520 (Figure 6A). Combining the two separate transcripts into a single contiguous transcript yields an ORF encoding a polypeptide of 7008 amino acids. A BLAST search in the TriTrypDB repository with this extended ORF sequence confirmed that the entire protein sequence is well conserved among other *Leishmania* species. In fact, an identical protein (7008 amino acids in length with 100% sequence identity) is annotated in the genome of *L. donovani* (strain LV9; gene ID: LdBPK.17.2.001090). Similarly, in *L. tropica* (strain L590), the orthologous protein (gene ID: LTRL590_170015100) is 7028 amino acids in length with a sequence identify of 91%; in *Leishmania turanica* (strain LEM423), the orthologous protein has 7032 amino acids (91% sequence identity), and the orthologue in *L. major* (strain Friedlin) has 7020 amino acids and a sequence identity of 90% (see Figure 6B for further details).

In several cases, Ribo-seq reads mapped to the 5′-UTR but clearly did not correspond to an extension of the annotated ORF, suggesting the possible presence of short upstream open reading frames (uORFs) in some transcripts. This class of short ORFs has been identified in mRNAs from different organisms; for instance, over 40% of mammalian mRNAs contain uORFs [28]. In general, uORFs are typically considered to be inhibitors of translation initiation at the downstream main CDSs, but their role may vary with developmental and environmental conditions [29,30,31,32]. To uncover plausible uORFs in *L. donovani* transcripts, we followed a quite stringent pipeline. The first step of the search strategy consisted of looking for potential ORFs (≥11 triplets in length) upstream of the primary annotated CDSs in the transcripts. Ribo-Seq reads were then mapped on these small ORFs, and those with more than five mapped reads were considered putative uORFs. In total, 304 putative uORFs were found in 264 different transcripts (see the Appendix A for a complete list). Figure 7 shows the position and details of a representative uORF, which is found in the 5′-UTR of the transcript LDHU3_32.2570T (encoding for a putative phospholipid/glycerol acyltransferase). A detailed analysis of the Ribo-Seq reads mapping to this region (Figure 7B) showed that the ribosome protected region is centered on the ATG initiation codon of the uORF (the protected region expands from 13 nucleotides upstream to 15 nucleotides downstream of the ATG). This finding suggests that this uORF would hinder the translation of the main ORF. Another similar uORF was previously described by Bifeld et al. [18] in the 5′-UTR of the HSP100-coding transcript using the *L. infantum* genome as a reference. Following the strategy described here, we identified the presence of the equivalent uORF in the corresponding transcript (LDHU3_29.1850T) in *L. donovani* (HU3) (see the Appendix A).

## 4. Conclusions

In this work, we reanalyzed the Ribo-Seq data generated by Bifeld et al. [18] to improve the *L. donovani* (HU3 strain) genome annotation in three ways. First, alignment of ribosome-protected fragments to the genome uncovered 71 new protein-coding sequences that had escaped previous algorithm-based methods, bringing the total number of annotated proteins in this strain up to 8463. Second, our analyses led to extension of the 5′ ends of 588 CDSs that were improperly truncated in the original automated annotations. Third, we uncovered eight instances in which single transcripts had been misannotated as two separate transcripts. These improvements, in combination with our recent assembly of the *L. donovani* (HU3 strain) genome and transcriptome [11], have resulted in the most complete and accurate annotation of an *L. donovani* genome to date. These data can serve as a reference for comparative genomics to improve the genome annotations for other *Leishmania* species. Our understanding of *Leishmania* biology and pathogenesis will be enhanced by our discovery of expanded and new coding sequences, which will increase the accuracy of transcriptomic and proteomic analyses. In addition, accurate definitions of CDSs are essential for functional genetic screening platforms. For example, we are currently generating a genome-scale inducible overexpression library encoding the majority of protein-coding genes in *L. donovani* (i.e., the *L. donovani* ORFeome) (P.A.Y. and J.M.R., unpublished). The updated annotations described herein will help ensure that PCR primers are designed to target the true boundaries of each ORF. The improved annotation will prove similarly useful for CRISPR/Cas9-mediated KO libraries and endogenous tagging strategies.

Finally, our analysis showed the presence of uORFs in 264 protein-coding transcripts in the *L. donovani* transcriptome, which is consistent with previous results found for *T. brucei* and *T. cruzi* [17,32]. These data provide a foundation for future studies investigating the role of uORFs in the translational control of gene expression in *L. donovani*.

## Figures and Tables

**Figure 1 genes-14-01637-f001:**
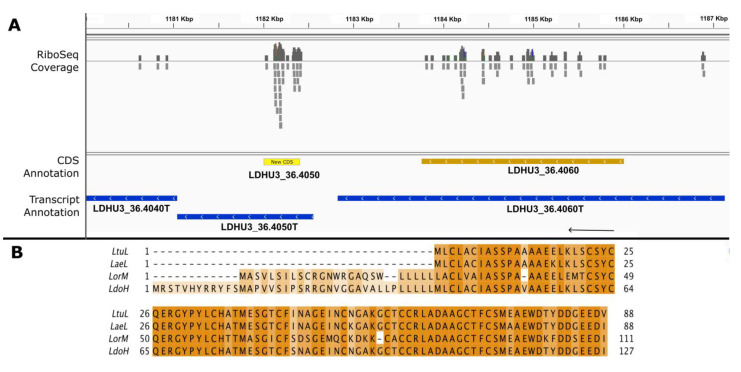
New CDS annotations in the transcript LDHU3_36.4050T. (**A**) Genomic view extracted via the IGV tool from Ribo-seq reads mapped on the *L. donovani* HU3 genome. Ribo-Seq coverage: a representation of the number of sequences aligned with this *L. donovani* genome subregion. CDS annotation and transcript annotation: previously annotated CDSs and transcript regions (dark colors) and new CDS annotations (yellow) based on Ribo-Seq data. (**B**) Sequence alignment of the new annotated protein with homologous proteins of other *Leishmania* species. LtuL: *L. turanica* strain LEM423, LTULEM423_360036100. LaeL: *L. aethiopica* strain L147, LAEL147_000839100. LorM: *L. orientalis* MHOM/TH/2014/LSCM4, LSCM4_00294. LdoH: *L. donovani* strain HU3, LDHU3_36.4050 (new CDS).

**Figure 2 genes-14-01637-f002:**
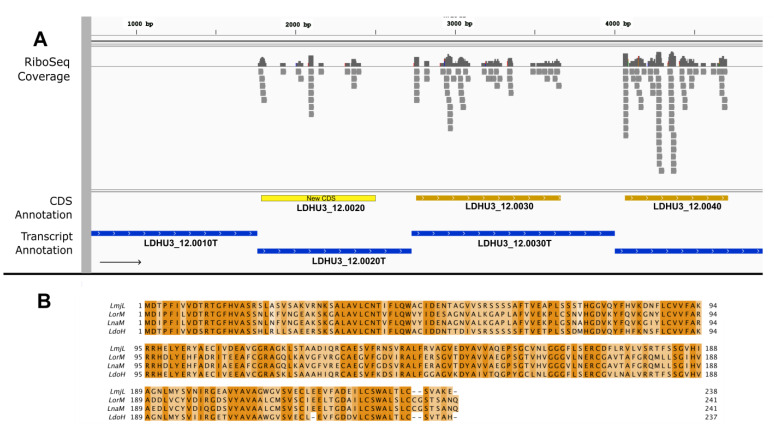
New CDS annotations in the transcript LDHU3_12.0020T. (**A**) Genomic view extracted using the IGV tool from Ribo-seq reads mapped on the *L. donovani* HU3 genome. Ribo-Seq coverage: representation of the number of sequences aligned with this *L. donovani* genome subregion. CDS annotation and transcript annotation: previously annotated CDSs and transcript regions (dark colors) and new CDS annotations (yellow). (**B**) Sequence alignment of the new annotated protein with homologous proteins of other *Leishmania* species. LmjL: *L. major* strain LV39c5, LMJLV39_120005000. LorM: *L. orientalis* MHOM/TH/2014/LSCM4 (LSCM4_07588). LnaM: *Leishmania* sp. Namibia MPRO/NA/1975/252/LV425, JIQ42_08299. LdoH: *L. donovani* strain HU3, LDHU3_12.0020 (new CDS).

**Figure 3 genes-14-01637-f003:**
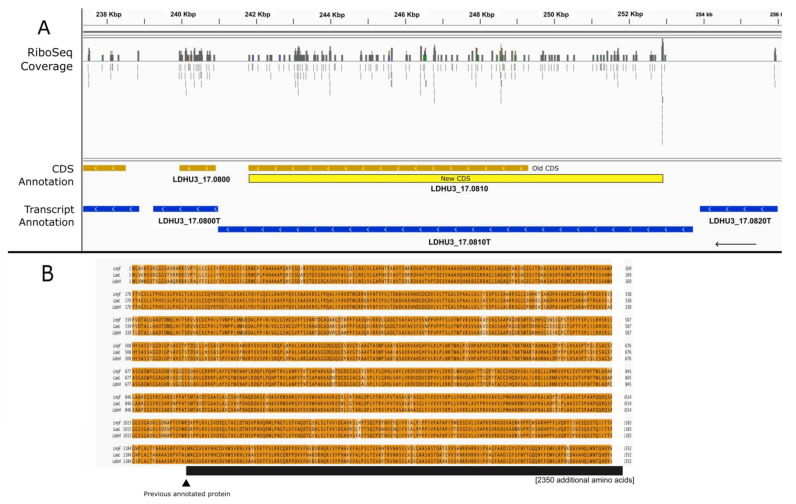
A 5′ extension of the CDS LDHU3_17.0810 based on ribosome profiling sequences (RiboSeq). (**A**) Genomic view extracted using the IGV tool from Ribo-seq reads mapped on the *L. donovani* HU3 genome. Color codes are as described in the legend for Figure 1. (**B**). Sequence alignment of the *L. donovani* HU3 extended sequence with homologous proteins annotated in other *Leishmania* genomes. LmjF: *L. major* strain Friedlin (LmjF.17.0470). LaeL: *L. aethiopica* L147 (LAEL147_000246200). LdoH: *L. donovani* HU3 (LDHU3_17.0810, extended CDS).

**Figure 4 genes-14-01637-f004:**
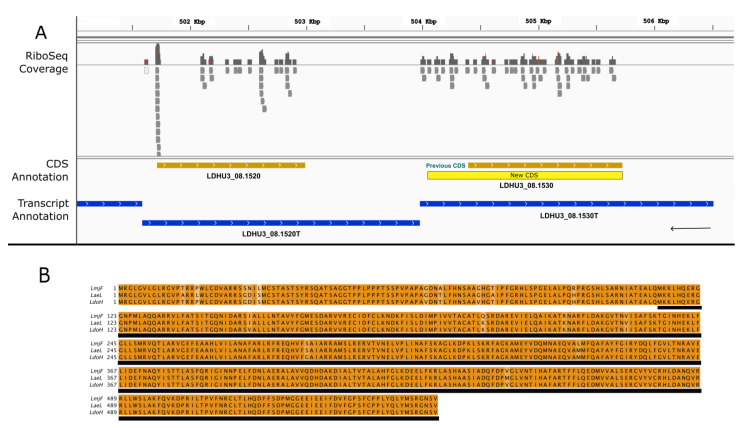
A 5′ extension of the CDS LDHU3_08.1530 based on ribosome profiling sequences (RiboSeq). (**A**) Genomic view extracted using the IGV tool from Ribo-seq reads mapped on the *L. donovani* HU3 genome. Color codes are as described in the legend for Figure 1. (**B**). Sequence alignment of extended protein with homologous proteins of other *Leishmania* species. LmjF: *L. major* strain Friedlin (LmjF.08.1170). LaeL: *L. aethiopica* L147 (LAEL147_000111700). LdoH: *L. donovani* strain HU3 (LDHU3_08.1530, extended CDS).

**Figure 5 genes-14-01637-f005:**
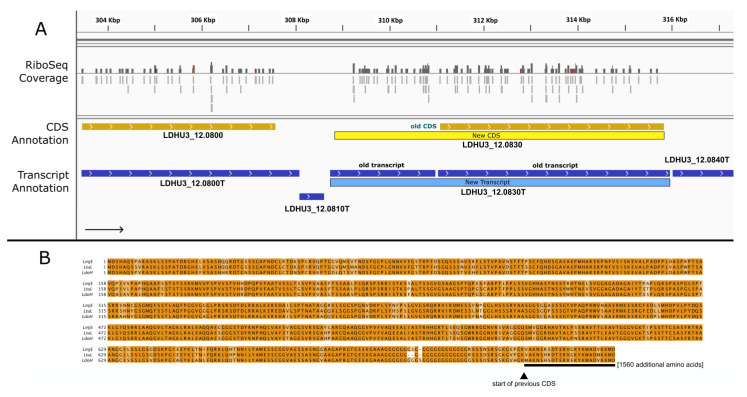
The fusion of previously annotated transcripts LDHU3_12.0820 and LDHU3_12.1830 to generate a new transcript (LDHU3_12.0830) with a 5′ extension in its CDS. (**A**) Genomic view extracted using an IGV tool from Ribo-seq reads mapped on the *L. donovani* HU3 genome. Color codes are as described in the legend for Figure 1. (**B**). Sequence alignment of the extended protein with homologous proteins of other *Leishmania* species. LmjS: *L. major* strain SD 75.1 (LMJSD75_120011200). LtuL: *L. turanica* strain LEM423 (LTULEM423_120011600). LdoH: *L. donovani* strain HU3 (LDHU3_12.0830, extended CDS).

**Figure 6 genes-14-01637-f006:**
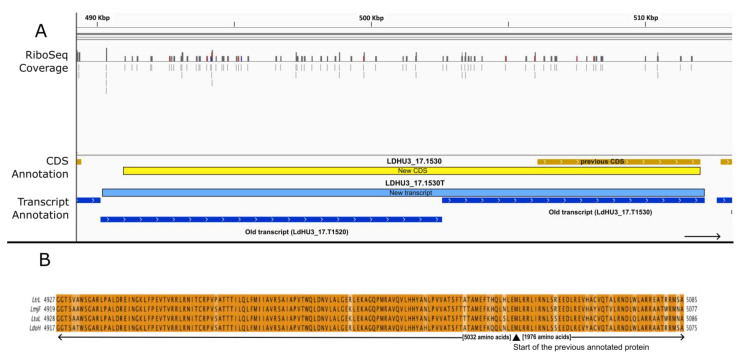
Fusion of previously annotated transcripts LDHU3_17.1520 and LDHU3_17.1530 to generate a new transcript (LDHU3_17.1530) with a 5′ extension in its CDS. (**A**) Genomic view extracted using the IGV tool from Ribo-seq reads mapped on the *L. donovani* HU3 genome. Color codes are as described in the legend for Figure 1. (**B**) Sequence alignment of the extended protein with homologous proteins of other *Leishmania* species. LtrL: *L. tropica* L590 (LTRL590_170015100). LmjF: *L. major* strain Friedlin (LmjF.17.0990). LtuL: *L. turanica* strain LEM423 (LTULEM423_170015400). LdoH: *L. donovani* strain HU3 (LDHU3_17.1530, extended CDS).

**Figure 7 genes-14-01637-f007:**
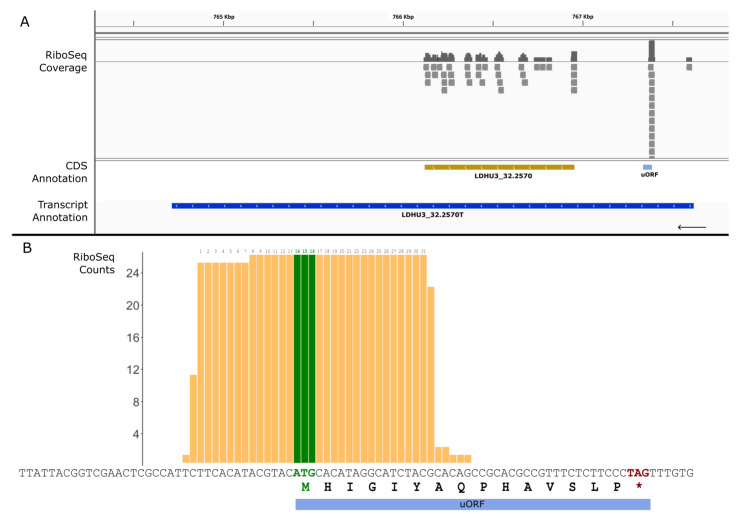
The identification of a possible uORF in the 5′-UTR of transcript LDHU3_32.2570. (**A**) Mapping of Ribo-Seq reads in the LDHU3_32.2570 gene locus. CDS (orange) and transcript (blue) annotations are depicted. In the 5′-UTR, a significant number of Ribo-Seq reads mapped on a region with a uORF. (**B**) Enlargement of the 5′-UTR where the Ribo-Seq reads mapped; the nucleotide sequence and the number of times a given nucleotide was covered by the reads are shown. The positions of the initiation (green) and termination (*, magenta) codons of the uORF are shown. Numbers on the bars indicate the nucleotides that are suggested to be protected by the stalled ribosome.

## Data Availability

See Section 2.6 of the Materials and Methods for detailed information.

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
