# Peer review of "Refinement of Leishmania donovani Genome Annotations in the Light of Ribosome-Protected mRNAs Fragments (Ribo-Seq Data)"

_genes, 2023, doi:10.3390/genes14081637_

Round 1

Reviewer 1 Report

The article has great relevance and brings important information (bioinformatics pipelines) to   to improve genome annotations of other organisms for which Ribo-Seq data are available. And also improvements to the complete and accurately annotated L. donovani genome and will enhance future studies in transcriptomics, proteomics and genetics.

It is worth mentioning that the authors of this work have expertise in data obtained by sequencing Ribo-Seq data and used data from annotations of a significant number of protein-coding genes in the genome of L. donovani (strain HU3), previously assembled and annotated by the group itself from a combination of short and long read sequencing data. And they identified uORFs in the 5' UTRs of a subset of L. donovani mRNAs, suggesting a potential conserved role for uORFs in the regulation of kinetoplastid translation.

Suggestion:

1. Conclusion could be restructured, as it is too long and contains information that has already been reported.

lines 387 and 388 could be deleted

Author Response

The article has great relevance and brings important information (bioinformatics pipelines) to   to improve genome annotations of other organisms for which Ribo-Seq data are available. And also improvements to the complete and accurately annotated L. donovani genome and will enhance future studies in transcriptomics, proteomics and genetics.

It is worth mentioning that the authors of this work have expertise in data obtained by sequencing Ribo-Seq data and used data from annotations of a significant number of protein-coding genes in the genome of L. donovani (strain HU3), previously assembled and annotated by the group itself from a combination of short and long read sequencing data. And they identified uORFs in the 5' UTRs of a subset of L. donovani mRNAs, suggesting a potential conserved role for uORFs in the regulation of kinetoplastid translation.

We appreciate these comments on our work. These are very encouraging to us. Many thanks!

Suggestion:

  1. Conclusion could be restructured, as it is too long and contains information that has already been reported.

lines 387 and 388 could be deleted

The mentioned sentence (lines 387 and 388) has been deleted. Furthermore, following the reviewer’s suggestion, the conclusion section has been shortened, trying to avoid reiterated information.

Reviewer 2 Report

The results of the analysis done in the present study improved the genome annotations of L. donovani parasite, enhancing future transcriptomics, proteomics and genetics studies. 

I think it can be accepted after minor revision:

In the introduction, at line 37: please add a more recent reference, like that at the following link: ": https://www.who.int/publications-detail-redirect/who-wer9635-401-419

Throughout the manuscript check the correct number of references cited, and possibly replace the reference with the correct number, as on page 3 at line 95, subtitute "Bifeld et l., 2018" with number 17.

I think some images need to be improved as sharpness and understanding. In Figures 1, 3, 4, 5, 7 the nucleotide scale is cut and often unreadable.

Check all the links mentioned in the manuscript for their correctness and indicate the last access.

Author Response

The results of the analysis done in the present study improved the genome annotations of L. donovani parasite, enhancing future transcriptomics, proteomics and genetics studies. 

I think it can be accepted after minor revision:

Thank you very much for your recommendation, and we appreciate the indications and suggestions. These have been incorporated in the revised manuscript as commented below.  

In the introduction, at line 37: please add a more recent reference, like that at the following link: ": https://www.who.int/publications-detail-redirect/who-wer9635-401-419

We have included the recommended reference, as it contains updated epidemiological data on leishmaniasis global epidemiology. In the revised manuscript, it has been incorporated as reference 2 [Ruiz-Postigo, J.A.; Saurabh, J.; Mikhailov, A.; Maia-Elkhoury, A.N.; Valadas, S.; Warusavithana, S.; Osman, M.; Lin, Z.; Beshah, A.; Yajima, A.; et al. Global leishmaniasis surveillance: 2019–2020, a baseline for the 2030 roadmap. Wkly. Epidemiol. Rec. 2021, 35, 401–419].

Throughout the manuscript check the correct number of references cited, and possibly replace the reference with the correct number, as on page 3 at line 95, subtitute "Bifeld et l., 2018" with number 17.

Following the reviewer’s suggestion and because most of the references has been renumbered, we have double checked the correct correspondence between the numbers and the appropriate references. Also, thanks for noticing us the incorrect manner in which the mentioned reference was included in the text.

I think some images need to be improved as sharpness and understanding. In Figures 1, 3, 4, 5, 7 the nucleotide scale is cut and often unreadable.

Following the reviewer’s recommendation, the layout of all the figures has been improved, and we have increased the font size to do readable some part of the figures.

Check all the links mentioned in the manuscript for their correctness and indicate the last access.

Following the reviewer’s recommendation, we have checked that all links are active. Accordingly, we have included the access date for all the links. Thanks!

Reviewer 3 Report

I want to forward the following comments about the paper. The manuscript is well-written and nicely organized. It details a well-designed and executed study.

Lines 94-95: The authors wrote that the description of the samples and the footprint methodology were described in another work. To help readers understand this work, it would be helpful to describe the samples and the methodology here briefly.

Line 149: There is an unnecessary space before the period (after the word "genome").

Avoid references in the Conclusions section and put your insight based on findings. 

Please double-check and improve the English.

Please double-check and improve the English.

Author Response

I want to forward the following comments about the paper. The manuscript is well-written and nicely organized. It details a well-designed and executed study.

We are grateful for this general comment on our work. Many thanks also for the comments and suggestions, which have been addressed as follows.

Lines 94-95: The authors wrote that the description of the samples and the footprint methodology were described in another work. To help readers understand this work, it would be helpful to describe the samples and the methodology here briefly.

Following the reviewer’s suggestion, we have added:

Briefly, promastigotes  of L. donovani  strain 1SR (MHOM/SD/62/1SR)  were cultured at 25°C in supplemented Medium 199 (Sigma-Aldrich) and harvested at a cell density of 6.7  × 106 ml-1. After cellular lysis and RNase I-treatment of the samples, isolated ribosomes (monosomes) were purified by sucrose gradient (10% to 50% [wt/vol]) centrifugation. Footprint RNA from the monosome fraction was isolated and used for library construction following the protocol of Ingolia et al. [12] with minor modifications. Finally, libraries were then sequenced via an Illumina NextSeq 500 system.

Line 149: There is an unnecessary space before the period (after the word "genome").

This has been corrected.

Avoid references in the Conclusions section and put your insight based on findings. 

 Most of the references were eliminated, and the section was extensively modified as requested by both Reviewer 3 and Reviewer 1.

Please double-check and improve the English.

Several minor edits have been made throughout the paper to improve the English grammar of the paper.

Reviewer 4 Report

This paper presents a significant advancement in improving the genome annotations of Leishmania donovani using Ribo-Seq data. The study successfully identified previously non-annotated protein-coding genes and enhanced the annotations of around 600 genes. Additionally, the authors provide evidence for the presence of small upstream open reading frames (uORFs) in many transcripts, suggesting a potential role in translational regulation of gene expression.

Overall, this paper provides a significant advancement in the field of genome annotations and opens up new possibilities for further research in L. donovani and other organisms. The work is well-structured and clearly demonstrates the importance of manual curation and integration of diverse data sources to achieve accurate gene annotations. It would be beneficial if the authors could briefly discuss the potential implications of the findings of newly uncovered protein coding sequences on our understanding of Leishmania biology and pathogenesis.

Author Response

This paper presents a significant advancement in improving the genome annotations of Leishmania donovani using Ribo-Seq data. The study successfully identified previously non-annotated protein-coding genes and enhanced the annotations of around 600 genes. Additionally, the authors provide evidence for the presence of small upstream open reading frames (uORFs) in many transcripts, suggesting a potential role in translational regulation of gene expression.

Overall, this paper provides a significant advancement in the field of genome annotations and opens up new possibilities for further research in L. donovani and other organisms. The work is well-structured and clearly demonstrates the importance of manual curation and integration of diverse data sources to achieve accurate gene annotations. It would be beneficial if the authors could briefly discuss the potential implications of the findings of newly uncovered protein coding sequences on our understanding of Leishmania biology and pathogenesis.

We appreciate these comments on our work. These are very encouraging to us. Many thanks!

Following the reviewer’s suggestion, we have included the following sentence (underlined) in the Conclusion section describing the potential implications of the newly uncovered proteins to improve our understanding of Leishmania biology and pathogenesis, within the context of the other likely contributions of this work to the field.

“These data can serve as a reference for comparative genomics to improve the genome annotations for other Leishmania species. Our understanding of Leishmania biology and pathogenesis will be enhanced by our discovery of expanded and new coding sequences, which will increase the accuracy of transcriptomic and proteomic analyses. In addition, accurate definition of CDSs is essential for functional genetic screening platforms.”
